# Benefits and Risks of Early Life Iron Supplementation

**DOI:** 10.3390/nu14204380

**Published:** 2022-10-19

**Authors:** Shasta A. McMillen, Richard Dean, Eileen Dihardja, Peng Ji, Bo Lönnerdal

**Affiliations:** Department of Nutrition, University of California, Davis, CA 95616, USA

**Keywords:** infant nutrition, iron supplement, iron deficiency anemia, growth, neurodevelopment, oxidative stress, trace mineral interactions, gut microbiome

## Abstract

Infants are frequently supplemented with iron to prevent iron deficiency, but iron supplements may have adverse effects on infant health. Although iron supplements can be highly effective at improving iron status and preventing iron deficiency anemia, iron may adversely affect growth and development, and may increase risk for certain infections. Several reviews exist in this area; however, none has fully summarized all reported outcomes of iron supplementation during infancy. In this review, we summarize the risks and benefits of iron supplementation as they have been reported in controlled studies and in relevant animal models. Additionally, we discuss the mechanisms that may underly beneficial and adverse effects.

## 1. Introduction

Iron is an essential trace element for human life: basic cellular reactions like energy production and DNA replication require iron, and in mammals, iron transports oxygen in the blood as hemoglobin. Insufficient iron intake to meet basic metabolic requirements leads to deficiency. Iron deficiency (ID) affects 10–40% of infants and causes approximately 50% of anemia cases worldwide [1,2,3].

Infants are especially susceptible to ID and iron deficiency anemia (IDA), both of which disrupt health and development [2,3,4,5], including adverse effects on long-term cognition and behavior. Once an infant becomes iron deficient, correcting iron status through dietary intervention prevents anemia, but may not correct disruptions to neurodevelopment and long-term cognitive development because critical phases of brain development occur during infancy [6,7].

Concern about the harms of ID has led to routine use of iron supplements to prevent ID [4,5]. Iron supplements, whether iron drops, multi-nutrient packets (MNPs), fortified formula, or fortified complementary foods, are effective at preventing or treating ID in most infants. Based on the success of iron supplements for preventing ID and IDA, the World Health Organization (WHO) recommends that iron supplements are provided to infants in populations where anemia prevalence exceeds 40% [4]. The same rationale backs the American Academy of Pediatrics’ (AAP) recommendations that exclusively breast-fed infants receive iron supplements beginning at 4 months, and that formula-fed infants receive iron-fortified formula [5].

The vast difference in iron intake—between the iron supplemented infant and the un-supplemented infant receiving only breast milk—is important but generally under-recognized. The Dietary Reference Intake (DRI) for iron for infants 0–6 months is 0.3 mg per day, based on the amount provided in breast milk [8]. For healthy infants born at term, liver iron stores in combination with the small amount provided in breast milk supports healthy growth and development up to 6 mo [9,10,11], but most fortified formulas in the USA contain 40× more iron than breast milk [12]. Even after accounting for differences in bioavailability between breast milk iron and formula iron, formula still provides around 7× more absorbable iron than breast milk. Furthermore, the AAP recommends 1 mg iron/kg daily supplementation for all exclusively or primarily breast-fed infants [5]. Following these recommendations, an iron-supplemented 5 kg infant would receive 17× more iron than what is provided by breast milk.

The WHO and AAP recommendations may lower the risk of ID, but infants with low risk of ID who receive iron supplements may be at risk of adverse effects, including disrupted growth and neurodevelopment, adverse nutrient interactions, and increased morbidity and mortality [13,14,15,16,17]. The mechanisms underlying these effects remain unclear and must be investigated for the risks of iron to be characterized; to predict which infants are the most vulnerable to which outcomes; and to improve efficacy and safety of iron provision [18,19].

Existing reviews in this area are of high quality [13,14,15,17,20], but they do not provide a full overview of outcomes that have been observed in controlled human and animal studies. The purpose of this review will be to summarize outcomes of iron supplementation between birth and 12 months in infants, as well as the corresponding developmental stage in animal models. With the goal of identifying the most promising directions for future iron intervention research, we discuss likely biological mechanisms underlying risks and benefits of iron supplementation during infancy.

## 2. Deficiency & Toxicity

Currently, the WHO recommends that infants age 6–23 months of age receive additional iron wherever anemia prevalence is estimated to be >40% [21]. Their recommendation is based on evidence from a meta-analysis of anemia outcomes [20]—part of a large systematic review by Pasricha et al.—which showed that iron was effective at increasing hemoglobin (*p* < 0.00001 for overall effect) and reducing risk of anemia for infants (0.61 relative risk; *p* < 0.00001 for overall effect) (see Appendix Figure A1) [20]. Reductions in anemia prevalence following iron provision are attributed to improvements in iron status, because risk of ID is typically also reduced [20]. Nevertheless, it is necessary to re-evaluate iron prophylaxis and its dose in infant populations—especially those living in areas of low risk of IDA—despite effective anemia prevention, because studies have reported adverse developmental outcomes of iron provision based on the current recommendations [22,23,24,25,26].

### 2.1. Defining Anemia

It is also necessary to re-evaluate the clinical definition of anemia for infants. The global cutoff for infants and children under 5 years (<110 g Hb/L) has been unchanged since it was defined in a 1968 WHO technical report on nutritional anemias [27,28]. The technical report cites infant data from two studies published in 1954 and 1959 [29,30], both of which included relatively small samples of infants (*n* = 237 and *n* = 129, respectively). In their current guide for assessing anemia [27] the WHO states their cutoff was “validated” by survey data collected in 1976–1980 during the National Health and Nutrition Examination Survey II (NHANES II), published by the United States Center for Disease Control & Prevention (CDC) in 1989 [27,31]. The CDC’s anemia cutoffs were derived from the 5th percentile Hb values, calculated from a “nationally representative sample” (*n* = 979) of “healthy” children 1–2 years old [31]. Notably, no Hb values were assessed in infants under 12 months old during this survey [31,32]. In summary, the current definition of anemia for infants is based on very limited evidence; therefore it is necessary to reconsider population-level iron recommendations designed around preventing anemia.

### 2.2. Defining Iron Deficiency

Hb defines anemia but is not a specific biomarker for iron status [2]. Specific iron biomarkers are serum ferritin (SF), serum iron, total iron binding capacity (TIBC), transferrin saturation, zinc protoporphyrin (ZPP) and soluble transferrin receptor (sTfR) [2]. Table 1 lists commonly used biomarkers as well as their cutoffs and their response to iron supplementation. Based on NHANES II data, the CDC recommends three biomarkers are used (ZPP, sTfR & SF), where at least two abnormal values would indicate ID [33]; however, often it is not feasible for clinicians to measure three biomarkers of iron in infants and young children. Instead, SF is used in combination with Hb to detect IDA. For anemic infants and children under 5 years of age, IDA is diagnosed when ferritin is <12 µg/L [34]. SF is a good biomarker for iron but also an acute phase protein that is elevated during systemic inflammation and thus may mask the presence of ID. Therefore, the WHO recommends a higher SF cutoff (<30 µg/L) to diagnose ID in infants in the presence of infection [34]. Researchers may assess inflammatory status alongside SF (e.g., C-reactive protein) to determine the validity of SF values; however, this method has not yet been standardized and the overall validity of SF as a biomarker for iron status during infancy remains uncertain [35].

Concrete evidence remains limited surrounding iron assessment and anemia diagnosis in infants, as well as the prevalence of infant ID or IDA. Thus, current practices for assessing iron status of infants and diagnosing ID or IDA rely on biomarker cutoffs that were defined by outdated, poor-quality evidence. Nevertheless, if one assumes these cutoffs are reliable, then there is evidence that: (1) there is high prevalence of IDA among toddlers 1–3 years of age [2]; (2) ID or IDA during infancy is associated with poorer developmental outcomes, particularly outcomes related to the nervous system and cognitive development [7]; (3) as stated above, controlled iron supplementation trials show reduced risk for ID and IDA [20]. However, whether iron supplementation improves and prevents poorer development outcomes is still unclear [15,20,36].

### 2.3. Iron Supplementation, Iron Status, & Hematology

There are inherent limitations to diagnosing ID and IDA during infancy, but there is good evidence that providing additional dietary iron will improve iron status. This is further supported by animal models: increased Hb, SF, transferrin saturation and serum iron, as well as increased liver iron concentration (a direct measure of iron stores) have been observed in swine, rats and mice. Studies in animal models support that Hb and iron biomarkers are elevated by iron supplementation but depend on baseline iron status as well as the dose, duration and form of iron supplementation.

Domesticated pigs are born without sufficient iron stores and must receive exogenous iron to prevent anemia (defined as <90 g Hb/L). Typically, 100–200 mg iron is administered to piglets during the first week of life as a single intramuscular or subcutaneous injection of iron dextran [37]. In one neonatal piglet study, non-supplemented piglets were severely anemic (mean Hb 72 g/L) by postnatal day (PD) 8, but piglets that had received an iron dextran injection or 5 days of oral iron had normal Hb levels (99 g/L and 100 g/L, respectively). SF, TIBC and serum iron, as well as spleen, liver, heart and kidney iron levels were also significantly elevated compared to non-supplemented piglets. Notably, greater iron loading in spleen, liver and kidney was observed in the iron dextran group compared to the oral iron group [38]. A small study from our group also found that non-supplemented piglets became severely anemic by PD 14, but iron dextran injections (100 mg iron) or oral iron (10 mg iron/kg BW as ferrous sulfate drops) prevented anemia at this age [39]. Similar results were reported in other pig studies [40,41,42,43]. Recently, in a larger and more robustly designed study where the control group received vehicle supplementation without additional iron, Hb was similar at PD 14 between control and iron-supplemented groups; anemia was observed at PD 35 only in the control group (82 g Hb/L), suggesting a long-term effect of iron supplementation [44]. The smaller effect of iron on Hb in the latter study [44] compared to the previous studies [38,39] may be explained by the differences in iron dose (1 vs. 10 or 15–20 mg iron/kg BW, respectively).

Rodent studies show that iron supplementation prior to weaning increases body iron levels, but effects on hematology are inconsistent [45,46,47,48,49,50,51,52,53,54]. Our group observed that in rat pups with ID, induced by a maternal low iron diet, iron supplementation corrected Hb and tissue iron levels [45]. Other studies show that hematopoiesis in rodents that were not ID at birth was either increased or unchanged by iron supplementation. Varying the iron dose produces variation in hematology and iron status outcomes, suggesting that iron intake levels affect hematopoiesis [39,41,44,46]. In summary, the extent to which iron supplementation improves iron stores and Hb in the pre-weanling animal depends on baseline iron status as well as the dose and type of iron administered.

### 2.4. Developmental Regulation of Iron

Homeostatic mechanisms control iron availability in early life, and postnatal growth necessitates rapid expansion of blood volume which increases demand for iron (Figure 1). Erythropoietin (EPO) is synthesized by the kidney in response to low oxygen or ID. Through endocrine signaling, EPO drives erythropoiesis and increases erythroferrone (ERFE) production in the bone marrow. ERFE signaling ensures that there is sufficient iron available for heme synthesis and RBC production by promoting iron absorption and increases circulating levels of iron, which is accomplished through suppression of hepcidin transcription in hepatocytes. Conversely, the iron regulatory hormone hepcidin is upregulated by bone morphogenic protein (BMP6) signaling in response to iron sensing by hepatic endothelial cells—in the absence of suppression by ERFE [55,56]. Hepcidin blocks ferroportin-mediated iron export in enterocytes, iron-storing hepatocytes, and spleen reticuloendothelial macrophages resulting in reduced iron circulation in the blood. This also leads to reduced iron absorption in the small intestine. The hepcidin-ferroportin axis serves as the systemic regulatory mechanism that prevents iron toxicity from dietary overexposure [57,58]. However, recent evidence suggests that this mechanism is not functionally mature in infants: Iron absorption is not well regulated in response to iron over-supplementation during the first year of life [9,10]. The same appears to be true for pre-weanling mice [59], rats [45,60] and piglets [39]. These animal studies show that intestinal ferroportin is hypo-responsive to hepcidin-induced degradation and permits elevated iron absorption during early development, despite substantial hepatic iron deposition [39,45,48,59,60]. This suggests infants are more vulnerable to iron overload.

### 2.5. Oxidative Stress Results from Iron Overload

Iron is a pro-oxidative element and iron overload in cells disrupts the oxidative balance by generating reactive oxygen species (ROS). Iron catalyzes the conversion of hydrogen peroxide into the highly oxidizing species hydroxyl radical. Iron overload thereby causes lipid, protein and DNA oxidation, which can ultimately result in cell death. This type of cell death caused by iron-induced lipid peroxidation and ROS accumulation is termed ferroptosis [61,62]. Mutations in the hepcidin-ferroportin pathway cause hereditary hemochromatosis (HH), an iron overload disease that demonstrates the pathological effects of iron toxicity. During HH, iron accumulates in the liver, where extreme iron overload initiates fibrosis, then cirrhosis and loss of liver function [63,64], eventually leading to complications and death if untreated [63,64]. In HH patients, liver fibrosis is believed to result from iron-induced oxidative stress [65]. Iron overload leads to extrahepatic iron loading, negatively affecting functions of other tissues. Thus, iron overload resulting from blunted regulation of iron absorption in early life may explain how excess iron can be harmful to development, but whether excess iron in early life causes tissue iron overload and oxidative stress remains to be investigated. Iron-toxicity injuries to developing organs like the liver would explain delays in growth and other adverse effects of iron supplementation in young children [10,13].

One double-blinded RCT investigated whether the amount of iron in formula alters blood markers of oxidative stress in infants [66]. Infants consuming 4 mg iron/L (as lactoferrin and FS) had greater plasma glutathione peroxidase activity (a marker of antioxidant activity) than those receiving more iron (6.9 mg iron/L as FS). The higher activity may have been due to higher levels of selenium in the 4 mg iron/L formulas, because selenium is a required component of glutathione peroxidase. When controlling for copper and selenium, there was no difference in glutathione peroxidase activity due to iron levels. Another RCT in Sweden and Honduras found that daily iron supplementation from 4–9 mo of age (at 1 mg/kg body weight, the current recommended dose) reduced plasma copper-zinc superoxide dismutase (SOD) activity, which is an antioxidant marker as well as an indicator of copper status [26].

Few other studies in human infants have reported effects on oxidative stress markers, but animal studies provide some insight. Nearly all iron supplementation studies that have measured oxidative stress in pre-weanling animal models have focused on oxidative stress in the CNS [39,48,49,53,67,68,69,70,71,72,73,74]. Our group observed no significant effect on hippocampal oxidative stress in pre-weanling rats or weanling piglets [39,48]. Dong et al. found that in piglets—which are born with ID—iron supplementation decreased expression of pro-inflammatory cytokines in the liver and spleen while increasing expression of genes involved in anti-oxidative activity [38]. Intriguingly, this effect was unique to the piglet group orally receiving ferrous glycine chelate iron, while the iron dextran injection group actually had increased expression of interleukin-1β and had no effect on antioxidant gene expression in the liver. Although both forms of iron had similar effects on Hb and SF, injection of iron dextran further increased hepatic and extrahepatic iron loading, and this likely contributed to the elevated inflammatory and oxidative stress markers [38]. Inconsistent oxidative stress effects have been observed in various brain regions in aging rodents following excess neonatal iron supplementation [49,53,67,69,70,71,72,73,74]. Few studies assessed tissue iron content or iron status when determining long-term oxidative stress effects of neonatal iron exposure. Kaur et al. observed increased oxidative stress in the substantia nigra (SN) of aged mice (12 mo) but not young adult mice (2 mo) following neonatal iron exposure, and this was associated with increased SN iron levels and reduced CNS motor circuit (nigrostriatal) activity [49]. Additional studies in human infants and animal models are necessary to understand how the pro-oxidative effects of iron might play a role in the growth and development outcomes of iron supplementation.

## 3. Growth & Development

### 3.1. Growth Effects of Dietary Iron Excess

Controlled studies have shown that iron supplementation of iron-replete infants negatively impacts their growth [22,24,75], but this effect has not been consistent in all studies [20,76]. A randomized placebo-controlled trial (RCT) reported iron supplementation from 4–9 mo reduced length-gain and head circumference-gain to 9 mo in Swedish infants who had low risk of ID [22]. A separate RCT in Indonesia found that iron provision reduced weight-for-age and length-for-age z-scores of iron-replete infants [24]. Another RCT in South East Asia found that iron supplementation from 6 to 12 months reduced length-for-age, but only in infants who had a healthy birth weight at baseline [75]. However, a more recent RCT from our group did not find any effects on growth metrics for healthy, full term Swedish infants from 6 weeks to 6 months [77]. It should be noted, though, that the previous three studies [22,24,75] all provided iron as drops, whereas the latter one provided iron in infant formula. A systematic review and meta-analysis of randomized controlled studies in children age 4–23 months reported negative effects of iron on weight and length gain [20], while another systematic review and meta-analysis of studies in children age 6–23 months did not find an effect on growth [76]. The difference in age of introduction of iron supplementation may explain this discrepancy–considerably more iron may be absorbed during 4–6 months of age when regulation of iron absorption is immature [9,10]. Yet, another comparable review and meta-analysis will investigate growth effects in iron-replete infants [36]. To date, there are insufficient studies to conclude the effect of iron supplementation on the growth of healthy, iron-replete infants. Nevertheless, the finding that iron is disruptive to growth in some cases demands further investigation into these effects.

### 3.2. Neurodevelopmental Outcomes of Iron Supplementation

The cognitive and behavioral effects of iron administration are also inconsistent [15]. Iron provision may prevent ID-related disruptions to nervous system development, but may be harmful to iron-replete infants, leading to long-term cognitive and behavioral deficits [78,79].

A well-powered, double-blind RCT conducted in Chile observed improved iron status and metrics of behavioral and social development in infants fed high-iron formula levels (12 vs. 2.3 mg iron/L as FS) from 6–12 mo of age. However, the pooling of breast-fed and formula-fed groups and the poor control of iron intake in this study muddles the interpretation of these results [80,81]. Moreover, despite exclusion of infants with IDA, ID may have been common at baseline. A follow-up study found increased response to reward, language abilities, and motor function in 10-year-olds who had been pooled into the high-iron group as infants. The authors did not report whether baseline iron status or estimated daily iron intake influenced behavioral outcomes of iron provision [82]. An additional follow-up study of this trial reported adverse cognitive and behavior effects in 16-year-olds who had received high-iron formula during infancy [79]. A small RCT in Canada found a positive effect of iron on Bayley’s scores of cognitive development [83], but iron intake from formula was poorly controlled and drop-out rates relatively high in this study. Another small RCT in Spain found that adding iron to cow’s milk improved the iron status of infants who were already iron-replete at baseline, but did not affect mental and psychomotor development metrics [84]. Thus, the impact of iron supplementation on long-term cognitive function is still unclear.

Ideally, supplement dose would be determined by an infant’s baseline iron status and optimized for healthy brain development, but this requires robust, well-powered studies. Unfortunately, few well-powered studies have measured baseline iron status or stratified results according to baseline iron status. One follow-up [78] of the same RCT above [80,81] found that after exclusion of anemic infants, baseline Hb predicted the effects of formula iron (12 vs. 2.3 mg/L) on cognitive development scores: infants with higher hemoglobin levels at baseline had poorer development scores at 10 years of age if they received high-iron formula, while infants with lower hemoglobin at baseline had improved development scores [78].

In a meta-analysis of RCTs, Pasricha et al. found that iron supplementation of all children aged 4–23 months did not affect Bayley’s mental or psychomotor development scores. Indeed, they observed a positive effect on Bayley scores when iron was provided to iron-deficient children, but stated there were insufficient well-powered studies to conclude whether iron provision is beneficial or harmful to iron-replete infants [20]. An upcoming systematic review from Hare et al. and meta-analysis may provide further insight on this matter [36]. Animal studies provide some compelling evidence that excess iron is harmful to brain development and leads to long-term cognitive and psychomotor deficits (discussed below); however, more human studies are needed to confirm these effects [15].

### 3.3. Mechanisms Underlying Neurodevelopmental Effects of Iron Supplementation

Iron is required not only for postnatal proliferation and differentiation of the central nervous system (CNS)—which begins prenatally and continues postnatally—but also for CNS-specific pathways, including neurotransmitter synthesis and myelination [85]. Brain regions with greater metabolic need for iron are programmed to import iron more rapidly than other regions. By this reasoning, such regions may permit excess iron loading and influence susceptibility to iron toxicity-induced oxidative stress. Oxidative stress damages CNS cells by triggering apoptosis, ferroptosis and necrosis.

The adult hippocampus is heavily myelinated, and the infant hippocampus requires relatively large amounts of iron because myelin synthesis is iron-demanding and peaks at this age. Myelin sheaths in the CNS are formed by oligodendrocytes, which wrap their myelin around neuronal axons, surrounding and insulating them to reduce axon resistance and accelerate signaling speed. Oligodendrocytes and their precursors must import and store sufficient iron for myelination, which is why ID leads to insufficient myelination. This may explain how ID during infancy leads to long-term cognitive and behavioral deficits; however, myelination is only one of many iron-demanding processes that take place in the CNS during the first year of life [86].

One study in pre-weanling rats observed that excess iron increased total iron content in the cortex, hippocampus, substantia nigra, thalamus, deep cerebellum and pons, but not in the striatum at PD 21. In contrast, supplying iron after weaning increased iron in the hippocampus and pons at PD 35, but not in other regions. Moreover, pre-weanling rats supplemented through PD 35 had elevated iron levels in the cortex, hippocampus, pons and superficial cerebellum [54]. These findings provide evidence that brain regions are differentially affected by iron supplementation.

Another study investigated how the timing of excess iron exposure affected oxidative stress in various brain regions. A gastric gavage of iron (10 mg iron/kg BW as ferrous succinate) was administered daily to rats PD 5–7, PD 10–12, PD 19–21 (pre-weaning), or PD 30–32 (post-weaning), and brain regions were assessed for oxidative stress at 3–5 mo of age (adulthood) [73]. They observed that pre-weanling iron exposure caused oxidative stress in the hippocampus, cortex and substantia nigra, suggesting a lasting effect of early life brain iron accumulation. Furthermore, CNS oxidative stress in this study was associated with impaired recognition memory. The hippocampus is part of the brain circuitry that encodes learning and memory—including spatial mapping and social cognition—and was also the region most consistently affected by oxidative stress in this study. The results from this study [73] are in agreement with recent studies from our group in piglets [39,87] and suggest that excess iron provision causes iron loading and oxidative stress in the hippocampus, associated with adverse effects on long-term cognitive function.

In a long-term animal study, oxidative stress was measured in brain regions of aging rats that were exposed to excess iron as neonates (oral gavage of 120 mg iron/kg BW as carbonyl iron). They found that pre-weanling iron overexposure elevated substantia nigra malondialdehyde (MDA) content (a marker for lipid peroxidation) and reduced glutathione content (a marker for antioxidant activity) at PD 400. These changes were associated with reduced dopamine neurotransmitter content in the striatum, as well as alterations in motor behavior, suggesting that excess iron in early life may lead to long-term dysfunction of the nigrostriatal pathway, a brain circuit involved in controlling movement, memory and response to reward [68]. Additional animal studies are needed to confirm these effects; however, these findings are congruent to cognitive and behavior effects in human infants [78].

## 4. Trace Mineral Interactions

### 4.1. Iron Deficiency May Mask Copper or Zinc Deficiency

Prolonged copper or zinc deficiency leads to iron deficiency. Copper is a required component of hephaestin and ceruloplasmin, which operate as co-transporters of iron [88]. Copper deficiency progressively diminishes the activity of these co-transporters, thereby reducing intestinal iron absorption, which causes iron deficiency. Similarly, zinc deficiency reduces iron absorption by suppressing the expression of the iron importer DMT1 and the iron exporter ferroportin [89,90,91].

In cases when ID derives from copper or zinc deficiency, iron supplementation would not be an effective treatment, as iron intake may be sufficient. Conversely, excess dietary trace mineral intake, such as excess iron, can affect absorption and metabolism of other trace minerals [92].

### 4.2. Iron Competes with Other Trace Minerals for Absorption & Metabolism

Excess iron may disrupt absorption and metabolism of other trace minerals. In a secondary analysis of a randomized, placebo-controlled trial, serum zinc decreased in infants after 6 months of iron supplementation, but only in infants that were iron-replete at baseline (6 months). However, a study in non-anemic Kenyan infants did not find an effect on serum zinc or zinc absorption with the addition of iron in micronutrient powder [93]. Copper-zinc superoxide (CuZnSOD) dismutase activity, a marker of copper status, was reduced in iron vs. placebo-supplemented infants at 9 months; however, no effect on serum copper was observed [26]. Insufficient research exists to ascertain that excess iron influences infant zinc and copper status, but similarities in biochemistry and pathways of absorption among iron, zinc, copper and manganese may explain how excess iron intake would disrupt trace mineral metabolism.

A pre-weanling rat supplementation study from our group demonstrated that tissue levels of zinc, copper, and manganese were altered by excess iron supplementation [46]. Pre-weanling rats with high iron intake had reduced liver copper levels and elevated levels of zinc in the liver, kidney, brain, and intestine compared to a vehicle control group. Prolonged supplementation with excess iron reduced zinc and copper levels in rat brains, reduced zinc and manganese in spleen tissue and caused elevated zinc in the liver. Lower levels of iron supplementation affected trace mineral levels to a lesser extent in the pre-weanling rats, suggesting that excess early life iron supplementation stimulates dysregulation of trace mineral metabolism. Nevertheless, additional studies are needed to determine how and when excess iron influences or disrupts trace mineral status in infants.

### 4.3. Trace Minerals and Oxidative Stress

The transporters DMT1, ZIP8, and ZIP14 import divalent metals including iron, copper, zinc, and manganese [45,94,95,96], therefore it is possible that high levels of iron may out-compete other divalent metals for import, and this may explain alterations in availability of these minerals in response to excess iron supplementation. Likely a secondary effect of cellular iron loading is the upregulation of trace metal binding and storage proteins such as copper-zinc SOD, manganese SOD, metallothioneins (MT) and ceruloplasmin (CP). MTs, CP, and copper-zinc or manganese SODs require these metals to function as ROS scavengers and their upregulation is induced by oxidative stress. By this reasoning iron loading may upregulate antioxidant, metal-binding proteins including MTs, SODs and CP by inducing oxidative stress. Since zinc and copper are needed for basic metabolism, growth and resistance to infection, disrupting their availability to growing organs and tissues would disrupt development and health [8]. However, it remains to be investigated if mineral interactions in the context of excess iron are linked to the adverse growth and development effects of excess iron.

## 5. Morbidity & Mortality

### 5.1. Iron Affects Morbidity & Mortality of Infants & Children

Approximately 90% of iron from FS, a common iron supplement, remains in the gastrointestinal (GI) tract until it is excreted [9]. The GI side effects of FS iron supplements are well-established: after pooling data from 43 studies, a meta-analysis of GI side effects of FS for adults estimated an 11% incidence rate for nausea, 12% for constipation and 8% for diarrhea [97]. Another systematic review and meta-analysis estimated that 1 in 3 adults who received FS supplementation experiences some adverse effects [98]. It seems likely that infants would be affected similarly, but this has not been fully investigated.

Iron provision has been associated with increased risk of diarrhea and respiratory infection in some studies [17,20]. If excess iron increases infection risk, this may explain how growth is negatively affected by iron supplementation in some cases [22,24,75]. A previous study from our group found that diarrhea frequency increased and growth was reduced for infants in Sweden and Honduras who had normal Hb levels at baseline, while the opposite was true for infants who were anemic at baseline [22]. For many studies reporting no effect of iron supplementation on diarrhea frequency, results are not stratified according to baseline iron status (provided that baseline iron status was measured in the study) [99,100,101,102,103,104]. One recent study found that infants who were treated with antibiotics experienced greater frequency of diarrhea if they were also receiving MNP with iron, as compared to infants who were treated with antibiotics while receiving MNP without iron, as part of a larger double-blinded RCT [25]. Increased iron availability in the gut may have increased the proliferation of diarrhea-causing *Clostridium difficile* in infants receiving high-iron formula and iron drops [105]. The bioavailability of iron (i.e., the extent to which iron is absorbed or passed through the gut) may be influenced significantly by intervention methodology: supplementation vs. fortification with iron, form of iron used, and timing of iron administration [9,11,17]. In their review of diarrhea outcomes [17], Ghanchi et al. suggested that supplementation may increase risk for diarrhea when compared to fortification; conversely, more expensive forms of iron (such as NaFeEDTA) may lower the risk for diarrhea compared to iron salts. Furthermore, common foods introduced as part of the complementary diet after 6 months of age influence iron absorption: grains, beans, and legumes contain indigestible phytates that reduce iron bioavailability, and citric or ascorbic acids in foods can augment iron absorption [106]. However, there are still an insufficient number of comparative studies to define the safest iron intervention methods for infants. In summary, current evidence suggests that baseline microbiota, iron status and iron intervention methodology are essential for predicting whether iron may increase morbidity in infants.

### 5.2. Gut Development & the Gut Microbiota

Alterations to the gut microbiota may contribute to GI side effects of iron supplements, as well as growth and development outcomes. Infancy is a critical period for symbiotic gut microbiota colonization and recent studies show that iron supplements alter the gut microbiota in ways that may be unfavorable to infant GI health [23,105,107,108]. Enteropathogens invade more easily during this age due to immature barrier function of the intestinal mucosa [109], leading to diarrhea or other infections [17,110]. Bacteria translocating across the mucosa trigger pro-inflammatory signaling, perhaps leading to diarrhea, both of which are likely to impair the nutrient absorption capacity of the GI tract [111]. Prolonged GI inflammation or diarrhea might therefore reduce an infant’s growth rate, suggesting GI effects are mechanistically related to adverse growth and development effects of excess iron.

An important aspect of development involves healthy colonization of the gut with commensal microbes, because the gut microbiota provides essential roles to their host’s health and development [112,113,114]. Besides maternal microbiota and birth method, the infant diet is the major determinant of gut colonization [114,115,116,117]. Breastfeeding and breast milk support healthy gut microbiota development by providing prebiotic oligosaccharides that preferentially craft the infant gut so that it is dominated by commensal *Bifidobacterium infantis*, which serves multiple health and development roles [118,119,120,121]. *Bifidobacterium infantis* has been shown to suppress the proliferation of pathogens and improve the integrity of the mucosal barrier, preventing inflammation and diarrhea [121]. Multiple studies have shown that iron reduces the abundance of commensal bacteria (including *Bifidobacterium infantis*) and elevates pathogen-associated bacteria [16,25,107]. Since commensal gut bacteria are so important for health and development, disrupting healthy colonization with commensals might explain some adverse outcomes of iron [108,112,122]. The gut microbiome of iron-replete infants may be more adversely affected by iron, but only one study has investigated gut microbiota outcomes in healthy, iron-replete infants [105].

A double-blind RCT of iron in micronutrient powder (MNP) given to Kenyan infants found increased abundance of *Clostridium* and *Escherichia/Shigella*–including increased pathogenic strains of *E. coli*–as well as elevated calprotectin, a measure of GI inflammation [23]. An additional robustly designed, double-blinded placebo controlled trial tested the effects of iron in MNP which was provided to 6-months old Kenyan infants for 3 months. In contrast to infants who received MNP without iron, infants who consumed MNP with iron had reduced abundance of commensal bacteria *Bifidobacterium* over time, while maintaining the abundance of *Escherichia* [107]. Another study from this group that was part of a large double-blind RCT in Kenya concluded that the addition of galacto-oligosaccharides (GOS) to the MNP with iron prevented its adverse effects on the microbiome. Despite the small sample size in this study, the results provide compelling evidence that iron adversely alters microbiome development by disruption of colonization by commensal bacteria [108]. A separate analysis that was part of this RCT followed gut microbiota changes and diarrhea outcomes in infants participating in the trial that had to be treated with antibiotics. Antibiotics were not as effective at suppressing the growth of enteropathogens or reducing diarrhea incidence in infants who were receiving MNP with iron, as compared to antibiotic-treated infants receiving MNP without iron [25]. These findings suggest that infants receiving iron supplements would be more susceptible to enteropathogens and have more diarrhea despite antibiotic treatment [35].

Disruptions to gut microbiota development lead to adverse effects on infant health, including alterations to GI development, metabolic signaling, brain development and immune system development [112,122]. Therefore, further studies are necessary to define how excess iron-induced alterations to gut microbiota development during infancy impact infant health and growth [13,16]. Considering that infant gut microbiota development is so important for overall development and that increased iron levels in the gut may cause adverse GI side effects and gut microbiota dysbiosis, it seems likely that the gut microbiota is involved in the adverse development effects of excess iron. Additional studies in animal models should characterize effects of excess iron on gut microbiota development and generate hypotheses about iron-induced alterations to the microbiota that may be causing adverse health and development outcomes.

## 6. Conclusions

Our conclusions are summarized graphically in Figure 2. Iron supplementation during infancy improves iron status, thereby reducing the risk of developing ID or IDA. However, the capacity of exogenous iron provision to disrupt health and development of otherwise healthy infants who are iron-replete is unclear because few existing studies have specifically measured iron status at baseline. Further, translationally optimized animal models are needed to investigate the mechanisms behind the adverse effects to infant health. Excess iron provision may delay growth and neurodevelopment and increase susceptibility to disease and infection, and it is likely that iron toxicity, mineral interactions and alterations in the gut microbiota are behind these outcomes.

## Figures and Tables

**Figure 1 nutrients-14-04380-f001:**
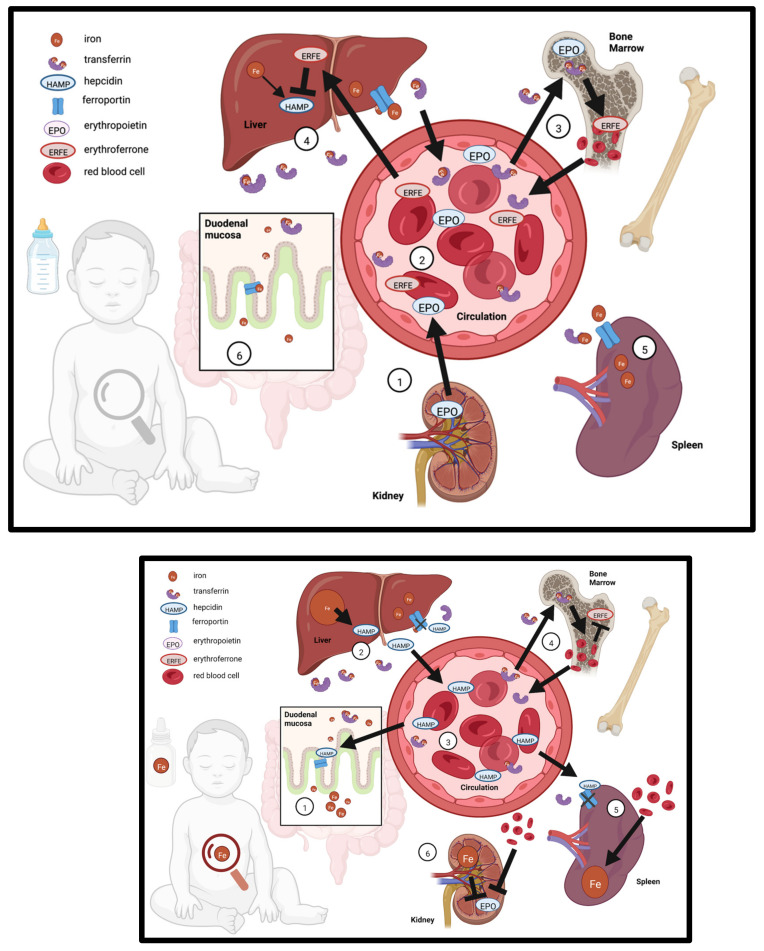
Iron regulation in the infant in response to iron supplementation. TOP—Iron regulation in the breast-fed infant in the absence of iron supplementation (**1**) The kidney secretes erythropoietin (EPO) to support the expansion of blood volume in response to low iron and low oxygen sensing; (**2**) EPO enters circulation and travels to the bone marrow, (**3**) where it drives erythropoiesis and secretion of erythroferrone (ERFE); (**4**) ERFE travels to the liver and suppresses hepcidin (HAMP) production, which allows for increased transferrin-bound iron in circulation; suppression of HAMP allows export of iron from the (**5**) spleen via ferroportin, as well as increased intestinal absorption of iron through ferroportin, both of which further increases iron in circulation. BOTTOM—Iron regulation in response to iron supplementation: (**1**) iron from supplementation is absorbed through the duodenal mucosa, is picked up by transferrin, and travels to the liver (**2**), increasing iron stores and up-regulating HAMP; (**3**) HAMP enters the circulation; (**4**) transferrin-bound iron supports increased erythropoiesis; in the spleen (**5**) erythrocytes are recycled but iron is sequestered because HAMP prevents export through ferroportin; (**6**) elevated iron and blood volume suppresses EPO production in the kidney.

**Figure 2 nutrients-14-04380-f002:**
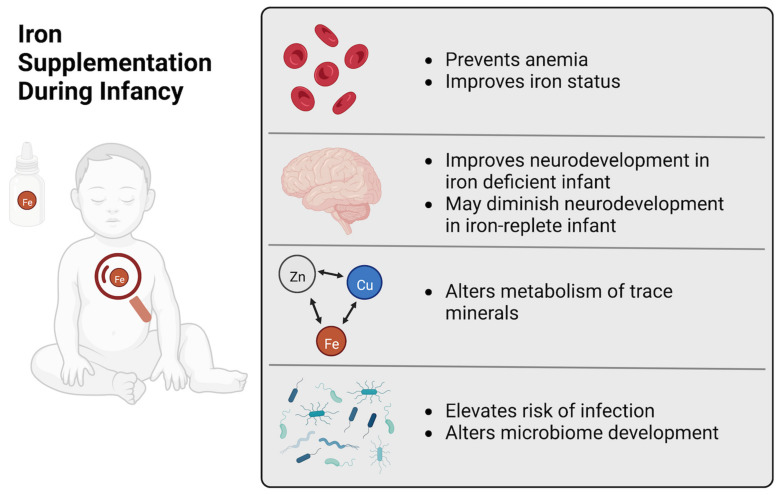
Iron Supplementation During Infancy. A graphical summary of the beneficial and adverse effects of iron supplementation of infants.

**Table 1 nutrients-14-04380-t001:** Common biomarkers for defining anemia and iron deficiency in infants.

Biomarker	Anemia Cutoff	Iron Deficiency Cutoff	Response to Iron Supplementation	References
Hemoglobin	<110 g/L	-	↑ or no change	[27,28,29,30,31,32]
Serum Ferritin ^1^	-	<12 µg/L	↑	[33,34,35]
Transferrin Saturation ^1^	-	<10%	↑	[32,33]
Zinc Protoporphyrin	-	80 μmol/mol heme	↓	[33]
Soluble Transferrin Receptor	-	8.3 mg/L	↓	[33]

^1^ Also elevated by inflammation [33]. ↑, Increased with iron supplementation; ↓, decreased with iron supplementation.

## Data Availability

The data presented in this study are not publicly available. The data are available on request from the corresponding author.

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
