# Peer review of "Benefits and Risks of Early Life Iron Supplementation"

_nutrients, 2022, doi:10.3390/nu14204380_

Round 1
Reviewer 1 Report
This is a very comprehensive review covering the pros and cons of Fe supplementation (e.g. for Fe-replete infants) during infancy. Especially helpful are the figures (Fig 1 the mechanisms; Fig 2, the summary) illustrating the authors' main points.
The authors clearly identify gaps in the reported evidence-base, which will help researchers possibly re-direct efforts, and recommending bodies (e.g. WHO, AAP) rethink "blanket recommendations" for Fe supplementation of infants birth-12 months of age.
Figure 1 is clear, however, to stand alone, may need more clarity (in the Illustration) to show that the Top infant is not receiving Fe in whatever they are being fed (e.g. there is no Food Visual in the top --what are they being fed? EBF? Non-Fe-supplemented formula?) vs. the iron-replete baby fed Fe-supplemented formula (represented clearly by the bottle with "Fe") in the bottom.
A Table would be helpful to summarize (and document) the Definitions of Anemia and ID for infants by Biomarker, and how (or if clear evidence exists that) each biomarker responds to Fe supplementation.
2.3: What about the different TYPES of Fe used in formulas and other products given to infants? Bioavailability of each of these? What about when other foods are introduced (e.g. by 4-6 mo) - how does this other dietary intake affect Fe absorption from supplements?
Author Response
[Comment 1] This is a very comprehensive review covering the pros and cons of Fe supplementation (e.g. for Fe-replete infants) during infancy. Especially helpful are the figures (Fig 1 the mechanisms; Fig 2, the summary) illustrating the authors' main points.
The authors clearly identify gaps in the reported evidence-base, which will help researchers possibly re-direct efforts, and recommending bodies (e.g. WHO, AAP) rethink "blanket recommendations" for Fe supplementation of infants birth-12 months of age.
[Response 1]
We thank the reviewer for taking the time to review our manuscript and provide feedback. We are pleased to receive this positive feedback from the reviewer!
[Comment 2] Figure 1 is clear, however, to stand alone, may need more clarity (in the Illustration) to show that the Top infant is not receiving Fe in whatever they are being fed (e.g. there is no Food Visual in the top --what are they being fed? EBF? Non-Fe-supplemented formula?) vs. the iron-replete baby fed Fe-supplemented formula (represented clearly by the bottle with "Fe") in the bottom.
[Response 2]
We have edited Figure 1 to show that the infant on the top is receiving milk without additional iron supplementation. The Figure 1 caption (Line 80) has been edited as follows:
“Figure 1. Iron regulation in the infant in response to iron supplementation. TOP—Iron regulation in the breast-fed infant in the absence of iron supplementation…”
[Comment 3] A Table would be helpful to summarize (and document) the Definitions of Anemia and ID for infants by Biomarker, and how (or if clear evidence exists that) each biomarker responds to Fe supplementation.
[Response 3]
We thank the reviewer for this suggestion. We have added Table 1 to the manuscript (Line 113):
Table 1. Common biomarkers for defining anemia and iron deficiency in infants.
Biomarker |
Anemia Cutoff |
Iron Deficiency Cutoff |
Response to Iron Supplementation |
References |
Hemoglobin |
< 110 g/L |
- |
↑ or no change |
[27-32] |
Serum Ferritin1 |
- |
< 12 µg/L |
↑ |
[33-35] |
Transferrin Saturation1 |
- |
< 10% |
↑ |
[32-33] |
Zinc Protoporphyrin |
- |
80 μmol/mol heme |
↓ |
[33] |
Soluble Transferrin Receptor |
- |
8.3 mg/L |
↓ |
[33] |
1Also elevated by inflammation [33]
We have also added the following statement to reference this Table in the text (Line 119).
“Table 1 lists commonly used biomarkers as well as their cutoffs and their response to iron supplementation.”
[Comment 4] 2.3: What about the different TYPES of Fe used in formulas and other products given to infants? Bioavailability of each of these? What about when other foods are introduced (e.g. by 4-6 mo) - how does this other dietary intake affect Fe absorption from supplements?
[Response 4]
We have added the following to the end of Section 5.1 (Line 436):
“The bioavailability of iron (i.e. the extent to which iron is absorbed or passed through the gut) may be influenced significantly by intervention methodology: supplementation vs. fortification with iron, form of iron used, and timing of iron administration [9,11,17]. In their review of diarrhea outcomes [17], Ghanchi et al. suggested that supplementation may increase risk for diarrhea when compared to fortification; conversely, more expensive forms of iron (such as NaFeEDTA) may lower the risk for diarrhea compared to iron salts. Furthermore, common foods introduced as part of the complementary diet after 6 months of age influence iron absorption: grains, beans, and legumes contain indigestible phytates that reduce iron bioavailability, and citric or ascorbic acids in foods can augment iron absorption [106]. However, there are still an insufficient number of comparative studies to define the safest iron intervention methods for infants. In summary, current evidence suggests that baseline microbiota, iron status and iron intervention methodology are essential for predicting whether iron may increase morbidity in infants.”
Reviewer 2 Report
Thank you for this excellent review of early life iron supplementation! It will be an important contribution to the literature and hopefully will stimulate more research in this area. The only comment this reviewer has is a request for separation by type font size between the Figure 1 legend and the subsequent manuscript text. The text beginning "2.1 Defining Anemia" should either be less indented or a different font size from the Figure 1 legend. Thank you again for an outstanding manuscript!
Author Response
We thank the reviewer for taking the time to review our manuscript and provide feedback. We are pleased to receive this very positive feedback from the reviewer!
The formating is determined by the Nutrients editors. We will work with the editors to better distinguish the legend and manuscript text around line 93.